# Psychological Distress and Associated Factors among Technical Intern Trainees in Japan: A Cross-Sectional Study

**DOI:** 10.3390/ijerph21080963

**Published:** 2024-07-23

**Authors:** Ei Thinzar Khin, Yuko Takeda, Kazunari Iwata, Shuhei Nishimoto

**Affiliations:** 1Department of Medical Education, Graduate School of Medicine, Juntendo University, Tokyo 113-8421, Japan; yu-takeda@juntendo.ac.jp; 2Department of Japanese Language and Literature, Faculty of Liberal Arts, University of the Sacred Heart, Tokyo 150-0012, Japan; iwata@u-sacred-heart.ac.jp; 3Innovative Organization for Human Resource Cultivation and Encouragement (iforce), Tokyo 111-0051, Japan; s.nishimoto@iforce.or.jp

**Keywords:** psychological distress, migrant health, mental health, occupational health, health-seeking behaviors, Japan, Asia

## Abstract

Japan is experiencing significant demographic shifts due to an aging and declining population. In 1993, the Japanese Government introduced the Technical Intern Training Program (TITP) to accept foreign national workers. While the number of technical intern trainees under this program has constantly increased, many of them face challenges in their daily lives, such as stress, health insecurities and limited access to healthcare. Therefore, we conducted a cross-sectional study to assess the mental well-being of technical intern trainees, focusing on psychological distress and its related factors. This study included 304 technical intern trainees from 12 prefectures in Japan, and was conducted from August 2021 to October 2021. We used self-administered questionnaires in the participants’ native languages to assess their sociodemographic conditions, health-related conditions such as health insecurities and healthcare-seeking behaviors, and psychological distress. The K6 Kessler Psychological Distress Scale was applied to evaluate the levels of psychological distress. Among the participants, 26.3% had moderate psychological distress and 2.3% had severe psychological distress. In addition, about 15% of the participants reported health insecurities and did not see a doctor despite wanting to. The multivariate model of logistic regression revealed significant associations between psychological distress and female gender (AOR 2.62, 95% CI 1.12–6.12), nationality other than Vietnamese (AOR 4.85, 95% CI 2.60–9.07), tough financial conditions (AOR 2.23, 95% CI 1.18–4.19), experiencing health insecurity (AOR 2.21, 95% CI 1.04–4.66) and the health behavior of refraining from seeking medical care (AOR 3.06, 95% CI 1.49–6.30). The top reasons for refraining from seeking medical care were the participants’ limited knowledge about the healthcare system in Japan and their worries about medical bills. These findings highlight the necessity to extend mental health support services, including counseling services, and share information about Japan’s healthcare system to supply medical services to the targeted technical intern trainee population.

## 1. Introduction

Like many other industrialized countries, Japan is experiencing significant demographic shifts characterized by a declining and aging population, and this evolving population structure has increased the demand for a diverse workforce in many industries [1]. In response to this demographic challenge and the need to maintain the labor force, the Japanese government initiated the Technical Intern Training Program (TITP) in 1993, a program further modified in 2016 [2]. These initiatives resulted from the progressive increase in foreign technical intern trainees across Japan. According to data from Japan’s Ministry of Health, Labor and Welfare, the reported number of foreign trainees more than doubled from 0.19 million in 2015 to 0.41 million in 2019 [3]. Technical intern trainees, much like other migrant workers worldwide, face multiple challenges and adapt to new and different environments [4]. Migrant workers globally encounter higher rates of occupational hazards, leading to poor health outcomes, injuries and fatalities. Health disparities arise due to cultural and language barriers, limited healthcare access, documentation status and the political climate [5,6]. Moreover, many studies have highlighted elevated levels of psychological distress and mental health issues among migrant populations, which can be attributed to factors such as isolation and discrimination [7].

In Japan, there are distinctions between technical intern trainees and other migrant workers. The primary objective of the government-supported program is to facilitate the development of emerging nations by imparting skills and expertise to these trainees [3]. However, over the years, some reports have characterized Japanese employers as utilizing it to fill low-wage positions and have yet to fully align with its intended objectives [8]. In 2017, 70.8% of workplaces (designated training providers) were found to have violated labor standards-related laws and regulations. The main violations contained issues related to (1) working hours (26.2%), (2) safety standards for machinery used (19.7%) and (3) payment of extra wages (15.8%) [9]. In recent years, there has been increased interest in the technical intern training program, resulting in a growing body of research investigating technical intern trainees’ health and working environment. While maintaining health is essential for a successful technical internship program in Japan, many reports have revealed considerable health concerns among technical intern trainees [10]. These include a rise in tuberculosis cases, an increase in occupational accidents, challenges in the environment and daily life in residences and an increase in mental health issues [11,12,13]. Despite strict labor standard-related laws and regulations and government measures aimed at preventing infectious diseases such as tuberculosis and addressing occupational accidents, there remains a necessity for comprehensive mental health support for technical intern trainees. On the other hand, addressing psychological distress and mental health issues among the migrant population presents challenges, notably arising from language barriers and the inaccessibility of the host country’s healthcare system [14]. This difficulty is also compounded by the mainly Asian origin of technical intern trainees in Japan, who often struggle with the high stigma surrounding mental health concerns in their countries of origin [15,16].

According to a literature review, it became evident that early detection of mild to moderate psychological distress is critical. Timely identification of such distress is essential for preempting its progression into more severe forms of mental illnesses, including major depressive disorders and suicidal tendencies [17]. Japan’s evolving standpoint on mental health awareness reflects a broader global trend of recognizing the importance of psychological well-being in the workplace [18,19]. Occupational health professionals have been at the forefront, advocating for the imposition of restrictions on working hours and the provision of timely consultations to address mental health concerns [20]. This changing activity underscores the urgency for migrants in Japan, who often work under challenging conditions, to have the means to assess and address their mental health status [21]. Recent research on migrant populations has indicated that foreign-born individuals are more likely to experience psychological distress [22,23], with prevalence rates varying depending on their country of origin [24]. Several studies have identified demographic and socioeconomic factors such as gender (specifically being female), economic status, length of stay in the host country [25,26] and language proficiency [24] as risk factors for psychological distress in migrant populations. Additionally, access to healthcare [26] and the availability of health information [27] have been found to have a negative correlation with psychological distress in these populations.

In this aspect, our research focused on technical intern trainees in Japan, a subpopulation group under the migrant category. This study included technical intern trainees from various nationalities across 12 different prefectures in Japan, representing different lines of work. This diversity enhanced a better understanding of psychological distress among technical intern trainees by considering different cultural backgrounds and work environments. Additionally, our research explored health insecurity and the healthcare-seeking behavior of technical intern trainees, areas that have yet to be extensively discussed in this population. On the basis of the literature review’s results, we hypothesized that psychological distress in technical intern trainees in Japan was associated with socioeconomic factors, health insecurities and healthcare-seeking behaviors. Specifically, our research posited that technical intern trainees from lower socioeconomic backgrounds, individuals with health insecurities and those who do not regularly seek healthcare services are more likely to experience heightened levels of psychological distress.

## 2. Materials and Methods

### 2.1. Study Objective

This study aimed to assess the psychological distress of technical intern trainees in Japan and to examine the factors associated with their psychological distress, with the goal of contributing to the development of a healthier environment for the technical intern trainees.

### 2.2. Study Design and Setting

This study was a cross-sectional, quantitative study. The study population was technical intern trainees in Japan.

A convenience sampling method was used for recruiting the participants, with study sites including Tokyo, Saitama, Kanagawa, Chiba, Shizuoka, Gunma, Nagano, Aichi, Mie, Gifu, Osaka and Fukuoka Prefectures. We conducted a questionnaire-based survey in collaboration with a supervisory organization from August 2021 to October 2021. A supervisory organization is a non-profit entity that manages the recruitment and supervision of technical intern trainees, provides advice and conducts audits on their employing companies. Before initiating the study, the supervisory organization explained the study’s protocol and objectives to each company, confirmed their understanding and obtained their cooperation.

### 2.3. Data Collection

During regular site visits to the companies and routine follow-ups with the technical intern trainees, staff members from the supervisory organization invited the trainees to participate in our study. The trainees were informed about the study objectives and oral consent was obtained before their voluntary participation.

The survey was conducted in multiple languages, with questionnaires translated into Vietnamese, Indonesian, Tagalog (Filipino), Thai, Mongolian and English. After completing the questionnaires, the participants sealed them in envelopes by themselves. The envelopes were collected on-site or later if immediate collection was not possible, and then sent to the research team for data entry.

No information on personal identification was collected during data collection. We assured participants that their survey responses would remain inaccessible to anyone from their host company or workplace.

### 2.4. Participants

During data collection, 348 questionnaires were distributed, and responses were received from all 348 participants, suggesting a 100% response rate. However, 304 respondents (87.4%) provided complete data, resulting in 304 valid responses available for data analysis.

### 2.5. Measurements

#### 2.5.1. Assessment of Psychological Distress

We used Kessler’s Psychological Distress Scale (K6) to assess psychological distress among the participants. Kessler’s Psychological Distress Scale, both the 10- (K10) and 6- (K6) item versions, were developed for the United States National Health Interview Survey (US-NHIS) to detect non-specific psychological distress for detection and evaluation purposes [28]. They are widely used due to their simplicity, clinical significance, self-administration, affordability and precise detection of severe mental disorders [29,30]. The K6 has good concordance with the larger K10 scale [31,32], and the Japanese version of the K6 showed a screening performance equivalent to the original English version [33].

The K6 is a self-reported survey asking about psychological distress symptoms experienced in the past 30 days, such as “feeling so depressed that nothing could cheer you up”. There is a five-category scale to record the responses (0 = all of the time, 1 = most of the time, 2 = some of the time, 3 = a little of the time and 4 = none of the time), thus producing a score range of 0–24.

The K6 has been shown to have cross-cultural reliability and validity, and the internal consistency (Cronbach’s alpha) for the K6 scale was 0.89 [33]. This scale was also used to access the psychological distress in migrant populations [34,35,36]. In our current study, the Cronbach’s alpha for the K6 scale was 0.79 for the Vietnamese and Tagalog (Filipino) translations, 0.67 for the Thai translation and 0.80 for the Indonesian translation.

#### 2.5.2. Assessment of Demographic Factors and Other Health Conditions

In addition to the K6, we collected data on the sociodemographic distribution and other health conditions of the participants. The demographic factors included age, gender, nationality, line of work, Japanese language skills, length of stay in Japan and financial situation. For the health conditions, we examined the body mass index (BMI) and a self-reported health condition, divided into categories such as “good”, “not so good” and “poor”. We also explored if there was a sense of insecurity or concern about health and any experience of not going to see a doctor though they initially wanted to. For the participants who had experienced not seeing a doctor initially even though they wanted to, the reasons were identified, for example, financial constraints, lack of time, cultural factors, etc.

### 2.6. Statistical Analysis

The distribution of the dependent variables and covariates was summarized according to the nature of the data. The characteristics of the participants are described in Table 1, and all the categorical variables are presented as the frequency (*n*) and percentage (%). Age and BMI were converted into categorical variables using standard cut-offs.

K6 scores of ≥5 and <13 were defined as moderate psychological distress, and K6 scores of ≥13 were defined as severe psychological distress [37,38]. As descriptive data, the psychological distress and other health-related conditions are described in Table 2.

In this study, we identified the dependent variable as the presence of moderate to severe psychological distress (K6 ≥ 5) and recoded it as binary data. We used a chi-squared test to compare the characteristics and health behaviors between the participants with no phycological distress (K6 < 5) and those with moderate to severe psychological distress (K6 ≥ 5). Then a binary logistic regression was conducted to determine the association between psychological distress and each covariate, choosing variables based on a *p*-value of <0.05 and their conceptual significance (Table 3).

The final model of adjusted logistic regression consisted of age, gender, nationality, line of work, stay in Japan (years), Japanese conversational skills, financial condition, experience of health insecurity and the experience of not going to see a doctor though initially intending to. Statistical significance was defined as a *p*-value of <0.05, and strong significance was defined as *p*-values of <0.01 and <0.001, with a 95% confidence interval.

All statistical analyses were performed with STATA version 17MP (Stata Corporation, College Station, TX, USA).

## 3. Results

### 3.1. Characteristics of the Participants

The study included 304 technical intern trainees. The characteristics of the participants are presented in Table 1.

The participants’ ages varied, with the youngest being 20 and the eldest 38, and the mean age was 27.1 ± 4.5 of standard deviation. Of these, 68.4% were between 20 and 29 years of age, and 31.6% were between 30 and above. Regarding gender, 60.5% of the total identified as female and 39.5% as male. Most participants were Vietnamese, which was 66.4% of the total. The remaining participants were from other Asian countries, including the Philippines (24.7%), Thailand (4%), Indonesia (4%) and others (0.9%).

Food production was the primary line of work (type of training) for a significant portion of the participants (59.6%). Others worked in construction (28.1%), nursing care (10.6%) and other fields (1.7%). About the length of stay in Japan, the majority of participants (over 70%) had been residing in Japan for at least 2 years. Additionally, approximately 90% of the participants demonstrated good conversational skills in Japanese, reflecting a commendable level of proficiency in the Japanese language. In terms of the financial situation, 24.7% experienced financial challenges ranging from a little to a very tough situation, while the majority reported being financially stable.

**Table 1 ijerph-21-00963-t001:** Characteristics of the participants.

	Frequency (*n*, %)
Total participants	304 (100)
Age, mean	27.1 (SD—4.5)
Age group (years)	
20–29	208 (68.4)
30–38	96 (31.6)
Gender	
Male	120 (39.5)
Female	184 (60.5)
Nationality	
Vietnam	202 (66.4)
Philippines	75 (24.7)
Thailand	12 (4.0)
Indonesia	12 (4.0)
Other	3 (0.9)
Line of work	
Food production	180 (59.2)
Construction	85 (28.0)
Nursing care	32 (10.5)
Others	7 (2.3)
Length of stay	
<1 year	27 (9.1)
≥1 year but <2 years	47 (15.8)
≥2 years but <3 years	155 (52.0)
≥3 years	69 (23.1)
Conversational skills in Japanese	
Hardly able to converse in Japanese	20 (6.6)
Enough for daily living	138 (45.4)
Work purposes without issues	144 (47.3)
Like a native	2 (0.7)
Financial situation	
A little to very tough	75 (24.7)
Okay	201 (66.1)
A little to significant surplus	28 (9.2)

Abbreviation: SD = standard deviation.

### 3.2. Psychological Distress and Health Condition of the Participants

Regarding the K6 score for psychological distress, the total K6 score of the participants ranged from 0 to 16 as described in Figure 1. Among the 304 participants, 217 participants (71.4%) had a K6 score of <5, indicating no psychological distress, while 80 participants (26.3%) had a K6 score of ≥5 and <13, which suggested moderate psychological distress, and 7 participants (2.3%) had a K6 score of ≥13, which suggested severe psychological distress (Table 2).

Most participants, comprising 83.1%, had a healthy BMI falling within the BMI = 18.5–24.9 category. This study assessed the self-reported health condition of the technical intern trainees. Participants were asked to categorize their health condition as “excellent”, “very good”, “good”, “not so good” or “poor.” For descriptive analysis, the data were categorized into “good to excellent” and “not so good/poor”. The results indicated that a majority of the participants (81.8%) reported their health condition as “good”, while 18.2% categorized their health as “not so good” or “poor”.

In the questionnaire, we also asked about participants’ behaviors of seeking medical care. A considerable 14.8% of the participants admitted experiencing a sense of insecurity or concern about their health. Moreover, 15.1% of the participants disclosed that they had expressed a desire to see a doctor but had refrained from doing so.

**Table 2 ijerph-21-00963-t002:** Psychological distress and other health-related conditions.

	Frequency (*n*, %)
Psychological distress	
No distress (K6 < 5)	217 (71.4)
Moderate distress (5 ≤ K6 < 13)	80 (26.3)
Severe distress (K6 ≥ 13)	7 (2.3)
BMI (kg/m^2^)	
<18.5	31 (10.2)
18.5–24.9	253 (83.2)
25.0–29.9	20 (6.6)
Self-reported health condition	
Good to excellent	249 (81.9)
Not so good/poor	55 (18.1)
Experienced a sense of insecurity or concern about health	
Yes	45 (14.8)
No	257 (85.2)
Wanted to see a doctor but ended up not going	
Yes	46 (15.1)
No	258 (84.9)

Abbreviation: BMI = body mass index.

### 3.3. Factors Associated with Moderate Psychological Distress

Table 3 shows the characteristics and health behaviors of the participants with or without psychological distress. There was a significant difference in gender, nationality, and financial status between those with psychological stress and those without psychological stress. Regarding health conditions, 23% of those with psychological stress had health insecurities, and 27.6% had refrained from seeking healthcare. These two factors also showed significant differences in those with and without psychological distress, as determined by the chi-squared test.

To analyze the factors associated with moderate to severe psychological distress among technical intern trainees in Japan, binary logistic regression analysis was used, and the results of the unadjusted and adjusted odds ratios with confidence intervals and *p*-values are described in Table 4.

In the crude analysis, the variables of female gender, having a nationality other than Vietnamese, experiencing tough financial conditions, feeling insecure about one’s health and refraining from seeking healthcare were all significantly associated with psychological distress. In the final model of adjusted logistic regression, the variables with significant *p*-values in the crude logistic analysis were adjusted by each other and for additional covariates, including line of work, length of stay in Japan, and Japanese conversational skills.

In the results of the adjusted logistic regression, females were associated with more chance of experiencing psychological distress than males (AOR 2.62, 95% CI 1.12–6.12, *p* < 0.05). Regarding nationality, intern trainees from the Philippines, Indonesia and Thailand had significantly higher odds of psychological distress (AOR 4.85, 95% CI 2.60–9.07, *p* < 0.001) than those from Vietnam. The participants with tough financial conditions were more likely to experience psychological distress (AOR 2.23, 95% CI 1.18–4.19, *p* <0.05). The participants who felt insecure about their health status (AOR 2.21, 95% CI 1.04–4.67, *p* < 0.05) and those who avoided seeking medical care (AOR 3.06, 95% CI 1.49–6.30, *p* < 0.01) were at a higher risk of developing psychological distress.

**Table 3 ijerph-21-00963-t003:** Characteristics and health behaviors of the participants with and without psychological distress.

	K6 < 5 (*n* = 217)	K6 ≥ 5 (*n* = 87)	*p*-Value
	*n* (%)	*n* (%)	
Age group (years)			
20–29	148 (68.2)	60 (69.0)	0.88
30–38	69 (31.8)	27 (31.0)	
Gender			
Male	95 (43.8)	25 (28.7)	<0.05
Female	122 (56.2)	62 (71.2)	
Nationality			
Vietnam	163 (75.1)	39 (44.8)	<0.001
Philippines	37 (17.0)	38 (43.7)	
Thailand	7 (3.2)	5 (5.7)	
Indonesia	7 (3.2)	5 (5.7)	
Other	3 (1.4)	0	
Line of work			
Food production	121 (55.8)	59 (67.8)	0.47
Construction	65 (30.0)	20 (23.0)	
Nursing care	25 (11.5)	7 (8.0)	
Others	6 (2.7)	1 (1.1)	
Length of stay			
Less than 2 years	56 (25.8)	18 (20.7)	0.35
2 years and more	161 (74.2)	69 (79.3)	
Conversational skills in Japanese			
Hardly able to converse in Japanese	11 (5.1)	9 (10.3)	0.14
Enough for daily living	96 (44.2)	42 (48.3)	
Work purposes without issues	110 (50.7)	36 (41.4)	
Like a native			
Financial situation			
A little to very tough	171 (78.8)	58 (66.7)	<0.05
Okay/surplus	46 (21.2)	29 (33.3)	
Experience of health insecurity			
Yes	25 (11.5)	20 (23.0)	<0.05
No	192 (88.5)	67 (77.0)	
Wanted to see a doctor but ended up not going			
Yes	22 (10.1)	24 (27.6)	<0.001
No	195 (89.9)	63 (72.4)	

Note: A chi-square test was used to compare categorical variables between participants with K6 < 5 and those with K6 ≥ 5.

**Table 4 ijerph-21-00963-t004:** Logistic regression analysis showing factors associated with moderate to severe psychological distress among the participants (*n* = 304).

Variable	Crude		Adjusted #	
	OR (95% CI)	*p*-Value	AOR (95% CI)	*p*-Value
Gender				
Male	Ref		Ref	
Female	1.93 (1.13–3.30)	0.016 *	2.62 (1.12–6.12)	0.026 *
Nationality				
Vietnamese	Ref		Ref	
Other	3.72 (2.20–6.26)	<0.001 ***	4.85 (2.60–9.07)	<0.001 ***
Financial condition				
Okay/surplus	Ref		Ref	
A little to very tough	1.86 (1.07–3.23)	0.028 *	2.23 (1.18–4.19)	0.013 *
Experience of health insecurity				
No	Ref		Ref	
Yes	2.30 (1.19–4.39)	0.012 *	2.21 (1.04–4.67)	0.038 *
Wanted to see a doctor but ended up not going				
No	Ref		Ref	
Yes	3.38 (1.77–6.43)	<0.001 ***	3.06 (1.49–6.30)	0.002 **

Abbreviations: OR = odds ratio, AOR = adjusted odds ratio, CI = confidence interval, Ref = Reference. Notes: # The adjusted logistic regression model consisted of age, gender, nationality, line of work, stay in Japan (years), Japanese conversational skills, financial condition, experience of health insecurity and the fact that the participant wanted to see a doctor but ended up not going. The goodness-of-fit tests for the multivariable logistic regression analysis were Pearson’s chi-squared test (*p* = 0.59) and the Hosmer–Lemeshow chi-squared test (*p* = 0.56); *p*-values: *** *p* < 0.001, ** *p* < 0.01, * *p* < 0.05.

### 3.4. Health-Seeking Behaviors and Barriers

We identified the factors influencing participants’ decision not to seek medical care despite their initial desire. Among the 46 participants who refrained from seeking medical care, the reasons for this choice were collected as multiple choice responses, represented in Figure 2 as a bar graph. Among the nine multiple choice answers for the reason, “knowing little about Japan’s medical system” (39.1%), “concerns about the medical bill” (32.6%) and “unable to communicate in Japanese” (30.4%) were the top three reasons.

## 4. Discussion

In this study, a considerable 28.5% of the total experienced moderate to severe psychological distress. Despite most of the trainees reporting good health conditions, about 15% of them had health insecurities and behaviors of refraining from healthcare. This important health situation and these health behaviors are significantly related to developing psychological stress among the participants. Moreover, the sociodemographic factors of females, non-Vietnamese trainees and those with a challenging financial status are possible risk factors for psychological distress.

Our findings have consolidated evidence of developing mental health issues in the female migrant population [25,39,40,41,42]. Of the 164 million migrant workers worldwide in 2017, 41.6% were female migrants, according to the 2018 ILO report [43]. Moreover, the Ministry of Health, Labor and Welfare of Japan estimated that the percentage of female migrant workers in Japan would increase by 25% between 2018 and 2022. [44]. Therefore, there is a solid case to prioritize mental health awareness initiatives and consultation services for the female migrant population in Japan.

The variable “nationality” had a significant association with psychological distress in our study. Specifically, participants originating from the Philippines, Thailand and Indonesia experienced psychological distress when compared with the Vietnamese participants. This result underscores the necessity of adopting a comprehensive approach to mental health research and the subsequent design of targeted interventions, which should contain a diverse spectrum of migrant populations from varying national backgrounds. According to the Japanese Ministry of Health, Labor, and Welfare (MHLW), as of the conclusion of October 2022, Vietnamese nationals constituted 25.4% of the overall foreign workforce in Japan, comprising a substantial 54.3% of the technical intern trainee demographic. Notably, there has been a growing body of research concerning the mental health and other health issues of the Vietnamese population in recent years [45,46,47]. However, mental health research for other nationalities in Japan remains limited. According to R. Bhopal et al., there are significant global disparities in the use of migration status and ethnicity as epidemiological variables [48], and it is crucial to include all data on ethnicity when conducting healthcare research [49]. As Japan becomes more culturally diverse, it is essential to give equal consideration to minority groups such as those from Indonesia, the Philippines and Thailand.

In financial circumstances, as per data from the Ministry of Justice, 50.0% of technical intern trainees identified their income as “low salary”, surpassing those with another residence status within the migrant population in Japan [44]. Many of these trainees travel to Japan to provide financial assistance to their families back in their home countries. Still, they often receive unsatisfactory pay for their labor [43]. Consequently, our study indicated that the combination of low income and financial hardship is closely associated with heightened psychological distress. These findings underscore the imperative for governmental authorities and training providers to be aware of these financial stressors and their potential impact on the mental well-being of this population.

In addition, this study has revealed significant associations among health insecurity, health-seeking behavior and psychological distress among technical intern trainees. In light of these findings, it is essential to create a supportive environment where trainees can comfortably discuss their health-related insecurities. Technical intern trainees who expressed unfulfilled desires for medical consultations exhibited a threefold higher risk of experiencing psychological distress in the results. Although many migrant populations worldwide face a higher likelihood of experiencing both physical and mental health challenges, they have limited access to healthcare services [26,50], and the same situation occurs for technical intern trainees in Japan [14,51]. The top reasons for bypassing medical consultations among the participants were “limited knowledge of Japan’s medical system” and “concerns about medical bills” in our study. Our findings underscore the importance of disseminating information about Japan’s healthcare system to technical intern trainees. Ideally, this provision should occur before arriving in Japan and commencing work at their training sites. It is advisable to provide comprehensive health information, in conjunction with guidance on preparedness for disasters and general living advice, to ensure the well-being of these intern trainees in their new environment.

This study has limitations. Firstly, convenience sampling may have introduced selection bias, potentially representing only some of the trainee population. However, we have addressed this by conducting a post hoc power analysis during the analysis of the data. Based on a final sample size of 304, this analysis showed a statistical power of 79.8% when the odds ratio was 2. This result indicated that our sample size was robust enough to detect psychological distress and draw valid conclusions about the population studied. Second, our study relied on self-reported questionaries, which could be affected by recall or social desirability bias, where participants may not accurately report their behaviors. Finally, our study did not include questions regarding healthcare conditions related to the COVID-19 pandemic. This information is critical, as the mental well-being and healthcare access of technical intern trainees in Japan may have been affected by the pandemic.

In the future, we recommend conducting longitudinal research to study the mental health of technical intern trainees, track significant changes over time and explore causal relationships among the variables, focusing on country of origin. It is also essential to investigate potential correlations between psychological distress and other factors such as acculturation, the social index and social support within this specific population. Furthermore, we advocate for qualitative interviews or mixed-method approaches to provide deeper insights into the health-seeking behaviors and challenges of access to healthcare faced by these intern trainees.

Recent reports have stated that the Japanese government will end the current Technical Intern Training system and establish a new one to ensure the sustainable development of human resources [52]. To develop this new system, we would like to propose a series of recommendations for the government and the stakeholders to enhance the working conditions for the technical intern trainees in Japan based on our research findings.

Counseling Services and Mental Health Support: Employers and supervisory organizations should implement culturally sensitive counseling services and inquiry counters or hotlines targeting female trainees. This support should be readily accessible and tailored to address the unique challenges foreign workers face in their daily lives and workplaces. Managers and supervisors should ensure that trainees are aware of and encouraged to use these mental health support services. They should provide regular check-ins to monitor their well-being and offer support as needed. The local government should promote awareness of the challenges faced by technical intern trainees to the community and support initiatives that aim to improve their mental health and overall well-being. Community members for their environment can play a crucial role in creating a welcoming and inclusive environment for these workers.

Educational Programs and Workshops about the Healthcare System: Sending organizations from overseas, employers from business enterprises and supervisory organizations should provide comprehensive educational programs such as workshops or informational sessions about Japan’s health system. These programs should be offered before the trainees’ arrival or shortly thereafter to help them navigate the healthcare system effectively. For example, employers should explain the health insurance system and other social insurance systems to the trainees to reduce their worries about medical expenses, and how they can access hospitals when needed. Managers and supervisors should facilitate participation in these programs and ensure that trainees have access to written materials, such as information handbooks in multiple languages, that explain the available healthcare resources.

Information on Accessible Healthcare Services: The available healthcare services such as clinics and hospitals in the residences should be provided for the technical intern trainees by employers or local government or community services when they start living in a new place. The supervisory organizations should participate in community engagement activities to foster a supportive environment for foreign nationals and encourage collaboration between local Japanese communities and foreign trainees to share information about health services and provide advice for easy access to healthcare services.

By involving all stakeholders in these initiatives, we can create a comprehensive support system that addresses the mental health needs of technical intern trainees in Japan. Our recommendations aim to improve their working conditions, enhance their overall well-being and contribute to the successful integration of foreign workers into Japanese society.

## 5. Conclusions

In conclusion, our research highlights the importance of customized interventions, such as targeting female trainees and providing support strategies in the new Technical Intern Trainee Program, to reduce psychological distress and improve the mental well-being of technical intern trainees. We recommend the introduction of counseling services, educational programs and informational handbooks in multiple languages to improve access to healthcare and change the health-seeking behavior of technical intern trainees. It is crucial for employers to be involved and for the community to participate in implementing these interventions for foreign national workers.

## Figures and Tables

**Figure 1 ijerph-21-00963-f001:**
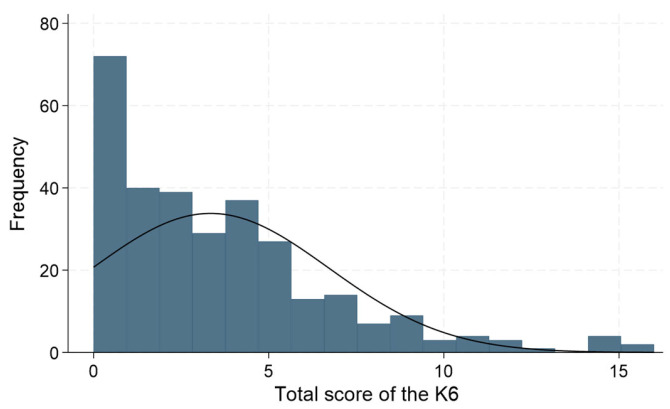
Distribution of levels of psychological distress (K6 scores) among technical intern trainees. Notes: The kurtosis was significant (*p* < 0.001), and the distribution was totally skewed to the left (*p* < 0.001). The Shapiro–Wilk test also showed a *p*-value of <0.001, indicating that the data significantly deviated from a normal distribution.

**Figure 2 ijerph-21-00963-f002:**
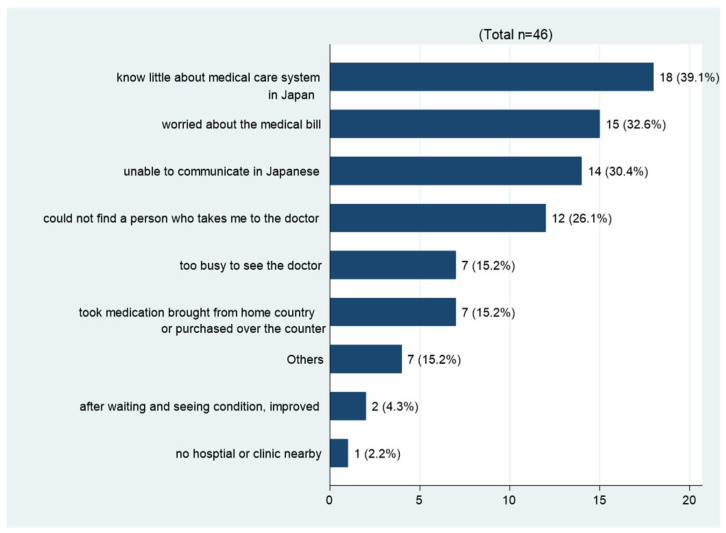
Reasons why the participants chose not to seek medical care despite their initial desire to.

## Data Availability

The raw data supporting the conclusions of this article will be made available by the authors on request.

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
