# Peer review of "Psychological Distress and Associated Factors among Technical Intern Trainees in Japan: A Cross-Sectional Study"

_ijerph, 2024, doi:10.3390/ijerph21080963_

Round 1

Reviewer 1 Report

Comments and Suggestions for Authors

Overall, this manuscript is technically sound. The components of the research are represented appropriately, as regard the title, introduction, methods, results and discussion, and conclusion. However, a few comments are needed to be cleared and fulfilled before it can be considered for publications.

Abstract

·        Line 22:  14.8% reported health insecurities: should start the sentence with letter not numbers (e.g. About forty or fifty percent……)

Materials and methods

·        Line 111-115: Mention the inclusion, exclusion criteria of the participants, how they were invited for participation in the study?

·        What about sample size, sampling technique used for recruiting the participants in the study?

·        In statistical analysis, what about normality test?

·        Line 168: choosing variables based on a p-value of 0.2 and conceptual significance (Table 3): there is error in writing this p value, the author means significant variables p < 0.05 chose for regression table??

Results:

·        Table 1: Length of stay “1 – 2 years, 2 – 3 years”: should be mutually exclusive e.g. 1< 2, 2-3 years

Author Response

Thank you very much for taking the time to review our manuscript and for the insightful comments. Please find our detailed responses in the attached PDF, along with the corresponding revisions and corrections highlighted in the re-submitted manuscript file.

We have also addressed typos and spelling errors. We hope this revision satisfactorily addresses each comment from the peer-review process.

We appreciate your valuable feedback and look forward to your further evaluation.

Reviewer 2 Report

Comments and Suggestions for Authors

The topic is intriguing, but for publication, the authors need to make significant improvements, such as:

  1. The Abstract needs enhancement. The authors mentioned that the study analyzes the well-being of..., but in the title and the study, it actually examines psychological distress (PD)...
  2. In the Introduction, the authors should include the novelty of the study and provide a brief description of each section analyzed.
  3. Before the Research Methodology, the authors should introduce a Literature Review section to discuss each variable, citing new sources from 2023-2024. They only included one source from 2024 and four from 2023, so additional sources are necessary. They should also formulate research hypotheses for each variable analyzed and discuss whether they were fulfilled in the Discussion section.
  4. The Research Methodology should clearly state the goal, objectives, and describe the methods used more explicitly.
  5. The Discussion section should include interpretations of the results, limitations, implications for employees, managers, and society, as well as suggestions for future research directions.
  6. The Conclusions need improvement.
  7. The References for the Literature Review should be updated with new sources for each analyzed variable influencing PD, as indicated in the title and the research.

The article is proposed with significant enhancements.

Author Response

Thank you very much for taking the time to review our manuscript and for the insightful comments. Please find our detailed responses and the corresponding revisions and corrections highlighted in the re-submitted manuscript file. 

Comment 1:
The Abstract needs enhancement. The authors mentioned that the study analyzes the well-being of..., but in the title and the study, it actually examines psychological distress (PD)...

Response 1: 
We wanted to mention the "mental" well-being of technical intern trainees. We revised it in the re-submitted manuscript. 

Comment 2:
In the Introduction, the authors should include the novelty of the study and provide a brief description of each section analyzed.

Response 2: 
In Line 96-103, we introduce the novelty of the study and some more detailed descriptions. 

Comment 3: 
Before the Research Methodology, the authors should introduce a Literature Review section to discuss each variable, citing new sources from 2023-2024. They only included one source from 2024 and four from 2023, so additional sources are necessary. They should also formulate research hypotheses for each variable analyzed and discuss whether they were fulfilled in the Discussion section.

Response 3: 
We updated the Literature Review section with new references from 2023-2024 related to the variables in our study. We also mentioned the hypothesis of the study. 
Please check the highlighted part for the revision. 

Comment 4: 
The Research Methodology should clearly state the goal, objectives, and describe the methods used more explicitly.

Response 4: 
Thank you for pointing this out. Based on your review and the other reviewers' comments, we revised the "Materials and Methods" section. 

Comment 5: 
The Discussion section should include interpretations of the results, limitations, implications for employees, managers, and society, as well as suggestions for future research directions.

Response 5: 
We have revised the re-submitted manuscript and hope our discussion covers the points you raised.

Comment 6: 
The Conclusions need improvement.

Response 6: 
We have made some edits to certain points.

Comment 7:
The References for the Literature Review should be updated with new sources for each analyzed variable influencing PD, as indicated in the title and the research.

Response 7:
Please see the updates in the literature review. 

We hope this revision satisfactorily addresses each comment from the peer-review process.
We appreciate your valuable feedback and look forward to your further evaluation.

Thank you. 

Reviewer 3 Report

Comments and Suggestions for Authors

First, I would like to thank you for the opportunity to read this work. I found this study interesting because it assessed the well-being of technical intern trainees from foreign countries who are working in Japan. This population is less studied; thus, this work contributes to the literature on the subject.

Below, I leave some comments, hoping they improve the paper.

Since the data was collected from August 2021 to October 2021, it will be important to address the COVID-19 pandemic issue.

This study has a 100% response rate. I wonder if participation was voluntary. Adding information concerning ethical issues and the anonymity guarantee will be important.

It is essential in the discussion section to identify the potential limitations of the study and suggest avenues for future research. 

Author Response

Thank you very much for taking the time to review our manuscript and for the insightful comments. We are pleased to hear that you found our study interesting and we appreciate your recognition of the importance of this work.

Please find our detailed responses to the comments, along with the corresponding revisions and corrections highlighted in the re-submitted manuscript file.

Comments 1:
Since the data was collected from August 2021 to October 2021, it will be important to address the COVID-19 pandemic issue.

Response 1:
Thank you for your valuable feedback. In our study, we did not include detailed questions regarding healthcare conditions related to the COVID-19 pandemic. The responses including the mental well-being condition and healthcare access of technical intern trainees in Japan which may indeed have been affected by the pandemic situation. We acknowledge this limitation and have addressed it in the limitations section of the manuscript.

Comments 2:
This study has a 100% response rate. I wonder if participation was voluntary. Adding information concerning ethical issues and the anonymity guarantee will be important.

Response 2:
We acknowledge the importance of ethical issues and anonymity.

We explained the study's objectives and invited the technical intern trainees to participate voluntarily. We also ensured the anonymity of all participants. We updated the data collection process and mentioned the anonymity of the participants in Lines 137-139.

During the data collection, we distributed the questionnaires to 348 participants, and all were collected back, so we assumed the response rate was 100%. This suggests that some trainees may have perceived that the survey was required to be answered since the questionnaires were distributed directly by members of their supervisory organization. On the other hand, even after the missing data imputation, we have 304 participants (87.4%), which may indicate their voluntary participation.

Comments 3:
It is essential in the discussion section to identify the potential limitations of the study and suggest avenues for future research.

Response 3:
We have revised the discussion section with the limitations of the study and the suggestions for future studies. Please see the highlighted part of the discussion section in the re-summitted manuscript.

We hope this revision satisfactorily addresses each comment from the peer-review process.
We look forward to your further evaluation.

Thank you. 

Reviewer 4 Report

Comments and Suggestions for Authors

The authors present a cross-sectional study of 304 technical intern trainees in Japan. The aim is to investigate psychological distress and other related factors in this sample. The results may be of interest as they provide insights into a current problem in the Japanese labour context.

The manuscript reports a well-written research paper that should be considered for publication. However, I suggest some minor revisions to improve the overall quality of the manuscript.

POINT 1. It is not clear in which language the survey was conducted. It seems that understanding the Japanese language could be a problem for the target population. Therefore, it is important that the authors indicate how they have accounted for the possibility that participants could not fully understand the items of the questionnaire (this, in fact, would have led to unreliable responses).

POINT 2. Lines 139-140: does the alpha value given come from the present study or from an earlier validation study?

POINT 3. The survey was conducted during the Covid period. Could this circumstance have had an impact on the participants’ responses?

POINT 4. Lines 245-246: there are some typos, so the sentence is not clear.

POINT 5. There was a multiple choice option for the reasons why technical intern trainees do not seek medical care. Was there also an additional open response so that participants could give possible explanations that were not considered by the authors? If this response option was not considered by the authors, it should be added to the limitations of the study.

POINT 6. Lines 283-284: this sentence refers to a period of time that has already passed (the increase should already have taken place).

POINT 7. The limitations of the study should also be reported (e.g. cross-sectional design, self-report measures…). Some directions for future research should also be given, such as the use of longitudinal models to measure significant changes over time and to test causal relationships between variables.

Author Response

Thank you very much for taking the time to review our manuscript and for the insightful comments. Please find our detailed responses, along with the corresponding revisions and corrections highlighted in the re-submitted manuscript file.

POINT 1:
It is not clear in which language the survey was conducted. It seems that understanding the Japanese language could be a problem for the target population. Therefore, it is important that the authors indicate how they have accounted for the possibility that participants could not fully understand the items of the questionnaire (this, in fact, would have led to unreliable responses).

Response 1:
We apologize for not specifying the language used in our study initially. Before the regular visit to the technical intern trainees, we contacted the companies in advance to determine the preferred languages of the technical intern trainees. The survey questionnaires were initially prepared in Japanese, and then translated into Vietnamese, Indonesian, Tagalog (Filipino), Thai, Mongolian and English. We hope this ensured that participants could fully understand the items in the questionnaire and enhancing the reliability of the responses.

We have updated the revised manuscript to include this important detail.  

Line 132-133: “The survey was conducted in multiple languages, with questionnaires translated into Vietnamese, Indonesian, Tagalog (Filipino), Thai, Mongolian, and English.”

POINT 2:
Lines 139-140: does the alpha value given come from the present study or from an earlier validation study?

Response 2:
This Cronbach's alpha, referenced with citation no. 27, was from a previous study, the 2008 World Mental Health Survey in Japan. We also calculated Cronbach's alpha for the K6 scale in our study and have added this in the revised manuscript.

Line 164-166: “In our current study, the Cronbach's alpha for the K6 scale was 0.79 for the Vietnamese and Tagalog (Filipino) translations, 0.67 for the Thai translation, and 0.80 for the Indonesian translation.”

POINT 3:
The survey was conducted during the Covid period. Could this circumstance have had an impact on the participants’ responses?

Response 3:
Thank you for your valuable feedback. In our study, we did not include detailed questions related to the COVID-19 pandemic. The responses including the mental well-being condition and healthcare access of technical intern trainees in Japan which may indeed have been affected by the pandemic situation. We acknowledge this limitation and have addressed it in the limitations section of the manuscript.

POINT 4:
Lines 245-246: there are some typos, so the sentence is not clear.

Response 4:
We have edited the writing in the revised version.

Line 271-274: “Regarding nationality, intern trainees from the Philippines, Indonesia, and Thailand had significantly higher odds of psychological distress (AOR 4.85, 95% CI 2.60 – 9.07, p < 0.001) than those from Vietnam. The participants with tough financial conditions were more likely to experience psychological distress (AOR 2.23, 95% CI 1.18 – 4.19, p <0.05).”

POINT 5:
There was a multiple choice option for the reasons why technical intern trainees do not seek medical care. Was there also an additional open response so that participants could give possible explanations that were not considered by the authors? If this response option was not considered by the authors, it should be added to the limitations of the study.

Response 5:
We included an open response option for participants who selected "others" as their reason for not seeking medical care, allowing them to specify any additional conditions or reasons not listed. However, the participants who selected this option did not provide further details in their questionnaires. We acknowledge that a qualitative interview conducted in the participants' native languages could provide deeper insights into these unspecified reasons. We have mentioned in our recommendations (Line 373-375) for future research.

POINT 6:
Lines 283-284: this sentence refers to a period of time that has already passed (the increase should already have taken place).

Response 6:
We have revised it to an easier-to-understand version.

Line 311-313 : “Moreover, the Ministry of Health, Labour and Welfare of Japan estimated that the percentage of female migrant workers in Japan would increase by 25% between 2018 and 2022.”

POINT 7:
The limitations of the study should also be reported (e.g. cross-sectional design, self-report measures…). Some directions for future research should also be given, such as the use of longitudinal models to measure significant changes over time and to test causal relationships between variables.

Response 7:
Thank you very much for pointing this out.
We have revised the discussion section with the limitations of the study and some suggestions for future studies based on your comments and other reviewers’ comments. Please see the highlighted part of the discussion section in the re-summitted manuscript.

We hope this revision satisfactorily addresses each comment from the peer-review process.

We appreciate your valuable feedback and look forward to your further evaluation.

Reviewer 5 Report

Comments and Suggestions for Authors

Dear Authors,

Psychological Distress and Associated Factors among Technological Intern Trainees in Japan: A Cross-sectional Study

Why was there a delay in publishing the research (2021)?

-Introduction :

The introduction is long and some paragraphs that are not related to the research can be shortened

The Statistical Analyses was appropriate and showed statistical relationships .the tables are appropriate and explain the results.

The discussion is Appropriate and acceptable

References and citations are appropriate, up-to-date and compatible with the purpose of the research

many thanks

Author Response

Thank you very much for taking the time to review our manuscript and for the insightful comments. We are pleased to hear your positive feedbacks and insightful suggestion.

Please find our detailed responses to the comments, along with the corresponding revisions and corrections highlighted in the re-submitted manuscript file.

Comment 1:

Why was there a delay in publishing the research (2021)?

Response 1:

We completed data management in 2022 and finalized data analysis by the end of 2022. During the following year 2023, we conducted online health consultations with participants and presented our findings at conferences and seminars in Japan to gather feedback. Following these activities, we dedicated time to meticulously revise and prepare our manuscript for submission to an international scientific journal, ensuring it met rigorous academic standards. This comprehensive process contributed to the timeline leading to publication.     

Comment 2:

Introduction

The introduction is long and some paragraphs that are not related to the research can be shortened

Response 2:

Based on your suggestion, we deleted some unnecessary parts in the Introduction section. However, as the other reviewer recommended, we had to add some updated references for the literature review and the research hypothesis.

We hope this revision satisfactorily addresses each comment from the peer-review process.

We look forward to your further evaluation.

Thank you. 

Round 2

Reviewer 2 Report

Comments and Suggestions for Authors

The authors improved some of the proposed measures, but they did not improve them all, so:

  1. Research hypotheses must be developed and new sources must be added for them.

2. Limitations were added, but not the implications for stakeholders; they must be highlighted using italic writing for each category.
Minor improvements are needed.

Author Response

Thank you for your feedback on our manuscript.

In our revised manuscript, we have included a hypothesis (Line 103 - 108).

"Based on the literature review results, we hypothesize that psychological distress in technical intern trainees in Japan is associated with socio-economic factors, health insecurities, and healthcare-seeking behaviors. Specifically, our research posits that the technical intern trainees, from lower socio-economic backgrounds, individuals with health insecurities, and those who do not regularly seek healthcare services are more likely to experience heightened levels of psychological distress."

We believe this hypothesis is well-supported by the literature we have cited (LINE 88 - 95).
These updated references and our comprehensive hypothesis adequately address the factors contributing to psychological distress among technical intern trainees. We are confident that these additions enhance the relevance and depth of our research.

In response to your comment 2 regarding the need to highlight implications for stakeholders, we have revised the discussion section to include detailed recommendations for various stakeholders, such as managers, employers, supervisory organizations, and the community. These implications are now clearly articulated and emphasized using italic writing for each category. (Line 381- 416)

Please see the revised version submitted on July 16. 

Thank you very much for your time.